# Retrieval-augmented Encoders for Extreme Multi-label Text Classification

## Abstract

Extreme multi-label classification (XMC) seeks to find relevant labels from an extremely large label collection for a given text input. To tackle such a vast label space, current state-of-the-art methods fall into two categories. The one-versus-all (OVA) method uses learnable label embeddings for each label, excelling at *memorization* (i.e., capturing detailed training signals for accurate head label prediction). In contrast, the dual-encoder (DE) model maps input and label text into a shared embedding space for better *generalization* (i.e., the capability of predicting tail labels with limited training data), but may fall short at memorization. To achieve generalization and memorization, existing XMC methods often combine DE and OVA models, which involves complex training pipelines. Inspired by the success of retrieval-augmented language models, we propose the Retrieval-augmented Encoders for XMC (RAE-XMC), a novel framework that equips a DE model with *retrieval-augmented* capability for efficient memorization without additional trainable parameter. During training, RAE-XMC is optimized by the contrastive loss over a knowledge memory that consists of both input instances and labels. During inference, given a test input, RAE-XMC retrieves the top-$K$ keys from the knowledge memory, and aggregates the corresponding values as the prediction scores. We showcase the effectiveness and efficiency of RAE-XMC on four public LF-XMC benchmarks. RAE-XMC not only advances the state-of-the-art (SOTA) method DEXML (Gupta et al., 2024), but also achieves more than 10x speedup on the largest LF-AmazonTitles-1.3M dataset under the same 8 A100 GPUs training environments. Our experiment code is available in the Supplementary Material.

## 1 Introduction

Many real-world applications, such as e-commerce dynamic search advertising (Prabhu et al., 2018) and semantic matching in product search (Chang et al., 2021), can be formulated as eXtreme Multi-label Classification (XMC) problems. These tasks involve retrieving relevant labels from a label collection of extremely large size $L$, typically under a tight latency budget. In such applications, $L$ can be millions or more, and the label space is often long-tailed, posing a significant challenge in designing XMC methods that are good at labels with varying frequencies (Dahiya et al., 2023a; Gupta et al., 2024). For head labels, it necessitates methods capable of *memorization*, which involves capturing all the detailed knowledge needed for accurate prediction. For tail labels, it requires *generalization*, as the model needs to capture general label representations that can be used to predict labels with few training instances.

Traditional one-versus-all (OVA) methods prioritize memorization via trainable label classifiers, yet struggle to generalize effectively across tail labels (Gupta et al., 2024). Recently, the emergence of pre-trained language models (LMs) has demonstrated strong generalization power, prompting the adoption of dual-encoder (DE) architectures to effectively leverage label text for zero-shot XMC tasks (Gupta et al., 2021; Xiong et al., 2022; Aggarwal et al., 2023). However, without extensive training, these models may not excel at memorization, suggesting a pure DE model is not sufficient to achieve good performance at head labels of XMC tasks.

To achieve both memorization and generalization power, competitive XMC approaches, such as NGAME$_{DE+OVA}$ (Dahiya et al., 2023a) and DEXA$_{DE+OVA}$ (Dahiya et al., 2023b), usually follow a complex two-stage solution. They first train a DE model that learns semantic embeddings between input and label text, which can generalize to unseen input queries and tail labels with text features.

Then the DE model is further augmented by trainable one-versus-all (OVA) label classifiers, which enhance the memorization capability of head labels with diverse intent of queries. Recently, the pioneering work, DEXML (Gupta et al., 2024), achieved SOTA results on several XMC benchmark datasets via a pure DE model, which significantly reduces the amount of trainable model parameters. While being a strong predictive model, DEXML requires extreme long training time to memorize the training signals of head labels, making it challenging to be applied to industrial applications.

In this paper, we introduce Retrieval-augmented Encoders for XMC (RAE-XMC), a novel framework that equips a dual-encoder (DE) model with the *retrieval-augmented* capability. RAE-XMC compensates for the poor memorization capabilities of DE and eliminates the need for precise matching between input instance and label embedding spaces, thereby significantly reducing the training difficulty of XMC. Specifically, RAE-XMC trains the DE model to contrast over a joint knowledge memory space that consists of label descriptions and input instances. During the inference, given a test instance, RAE-XMC first retrieves the top $b$ keys from the knowledge memory. RAE-XMC then generates predictions by aggregating the values (i.e., labels) of these keys based on their scores. The contributions of this paper are threefold as follows:

- We introduce RAE-XMC, a novel retrieval-augmented framework for XMC problems, which enhances the underlying DE model with better memorization capability. Specifically, RAE-XMC offers controllable trade-off between memorization and generalization, namely the performance of head and tail label segments, respectively (cf., Section 5.3).
- RAE-XMC significantly reduces the training difficulty for XMC problems in two ways. Firstly, it eliminates the need for training OVA label classifiers to memorize intricate complex patterns in head labels. Secondly, it alleviates meticulously training of the DE model to match label and input instance spaces (cf., Section 5.6).
- We conducted extensive experiments to showcase the effectiveness and efficiency of RAE-XMC. Our proposed approach not only advances the current SOTA results from DEXML (Gupta et al., 2024), but also comes with significantly less training time, achieving more than 10x speedup on the largest LF-AmazonTitles-1.3M dataset (cf., Section 5.2).

## 2 BACKGROUND MATERIAL

We study the eXtreme Multi-label Text Classification (XMC) problem with label text. Consider a training set of $N$ examples $\mathcal{D} = \{(x_i, \mathbf{y}_i)\}_{i=1}^N$, where $x_i \in \mathcal{X}$ is the $i$-th input instance and $\mathbf{y}_i \in \{0,1\}^L$ is the one-hot label vector with $y_{i,\ell} = 1$ indicating that the label $\ell$ is relevant to the input instance $x_i$. Depending on the availability of label text, $\ell$ denotes either a label index or label text, respectively. The label space $\mathcal{L}$ consists of $|\mathcal{L}| = L$ labels. The goal of XMC is to learn a scoring function $s_\theta : \mathcal{X} \times \mathcal{L} \to \mathbb{R}$, parameterized by model parameters $\theta$, such that $s_\theta(x, \ell)$ indicates the relevance between input instance $x$ and label $\ell$.

**Model Parametrization.** Based on the availability of label text features, there are two model families for the XMC task: the One-versus-all (OVA) classifier and the Dual-Encoder (DE) model. The OVA classifier, which ignores the label text features, consists of an input text encoder $f_\phi : \mathcal{X} \to \mathbb{R}^d$ and a classification matrix $\mathbf{W} \in \mathbb{R}^{L \times d}$. The scoring function of the OVA classifier then becomes $s_\theta(x, \ell) = \langle \mathbf{x}, \mathbf{w}_\ell \rangle$ where $\mathbf{x} = f_\phi(x)$ is the input query embedding and $\mathbf{w}_\ell$ (the $\ell$th row of $\mathbf{W}$) is the classification vector for label $\ell$. For the OVA classifier, the trainable parameter $\theta = [\phi; \mathbf{W}]$ scales linearly to the size of label space $L$, which can be large and difficult to train.

In contrast, our paper studies the DE model, where the scoring function is defined as $s_\theta(x, \ell) = \langle \mathbf{x}, \mathbf{z}_\ell \rangle = \langle f_\phi(x), h_\psi(\ell) \rangle$, with $f_\phi : \mathcal{X} \to \mathbb{R}^d$ as the input text encoder and $h_\psi : \mathcal{L} \to \mathbb{R}^d$ as the label text encoder. Compared to the OVA classifier, the trainable parameters of the DE model, $\theta = [\phi; \psi]$, are often much smaller since they do not scale with the size of the label space.

**Learning.** Given a training set $\mathcal{D}$, the scorer $s_\theta$ can be learned by minimizing some surrogate loss functions for some metrics of interest (e.g., Precision@k and Recall@k). One popular choice is the one-versus-all (OVA) reduction (Dembczynski et al., 2010) to $L$ binary classification tasks and employs binary cross-entropy (BCE) loss function. On the other hand, one can employ a multi-label to multi-class reduction (Menon et al., 2019) and invoke the Softmax cross-entropy loss:

$$J(x, \mathbf{y}; s_\theta) = -\sum_{\ell \in [L]} y_\ell \cdot \log\left(\frac{\exp\left(s_\theta(x, \ell)/\tau\right)}{\sum_{\ell' \in [L]} \exp\left(s_\theta(x, \ell')/\tau\right)}\right). \tag{1}$$

In practice, various negative sampling techniques are used to approximate the partition function for the Softmax cross-entropy loss, such as within batch negatives (Chang et al., 2020a; Karpukhin et al., 2020) and hard negative mining (Xiong et al., 2021; Dahiya et al., 2023a).

**Inference.**   Given a test query embedding $\mathbf{q} = f_\phi(q) \in \mathbb{R}^d$ and the offline pre-computed label embedding matrix $\mathbf{Z} \in \mathbb{R}^{L \times d}$, we aim to retrieve the $k$ most relevant labels from $\mathbf{Z}$ in real-time (low inference latency), which is also known as the Maximum Inner Product Search (MIPS) problem. Exact inference of MIPS requires $\mathcal{O}(L)$ time complexity, which is prohibited for XMC tasks where $L$ can be millions or more. Thus, practitioners leverage Approximate Nearest Neighbor Search (ANN) methods to approximately solve it in time sub-linear to the size of the label space $L$.

Conventional XMC methods with OVA classifiers learn a tree-based (Prabhu et al., 2018; Zhang et al., 2021; Yu et al., 2022) ANN index or graph-based ANN index (Liu et al., 2021; Gupta et al., 2022) at the training stage, and deploy it for fast retrieving top $k$ labels in $\mathcal{O}(\log L)$. In contrast, DE models (Dahiya et al., 2023b; Gupta et al., 2024) encode the label embedding matrix $\mathbf{Z}$, and apply existing ANN solvers (e.g., Faiss (Johnson et al., 2019), ScaNN (Guo et al., 2020), HNSWLib (Malkov & Yashunin, 2018)) to build the ANN index and achieve fast retrieval in $\mathcal{O}(\log L)$ time.

## 3 PROPOSED METHOD: RAE-XMC

### 3.1 OVERALL FRAMEWORK

The performance of Dual-Encoder (DE) methods for XMC is often hindered by the inherent challenge of *memorization*, requiring the encoding of every detail to effectively predict head labels. Inspired by the recent success of retrieval-augmented language models (Khandelwal et al., 2020; Guu et al., 2020; Izacard et al., 2023; Borgeaud et al., 2022),which have demonstrated remarkable efficacy in leveraging external knowledge to reduce the necessity of encoding all the factual knowledge, we propose the RAE-XMC framework. Rather than relying solely on the encoding capabilities of DE models, which may struggle to encapsulate all relevant information, RAE-XMC integrates retrieval mechanisms to augment the inference process.

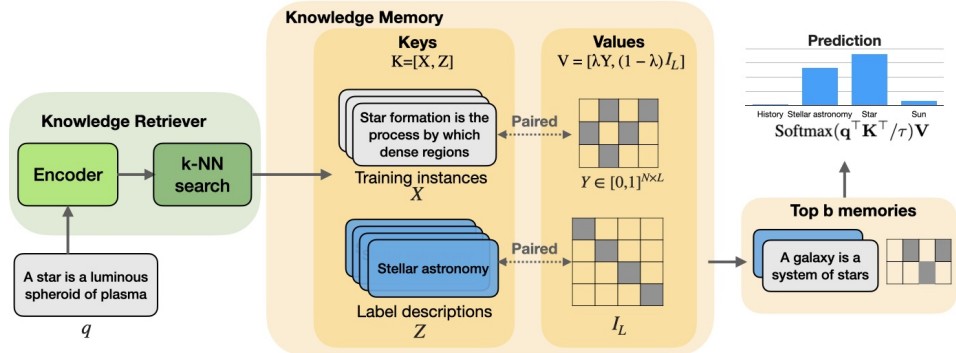

Figure 1: The proposed RAE-XMC framework. The knowledge retriever consists of an encoder and a $k$-NN searcher, which retrieves the top $b$ (key, value) pairs from a joint knowledge memory. The key consists of embeddings of training instances and label text descriptions, while the value consists of their corresponding one-hot label vectors. A lightweight predictor then combines the labels based on their scores to generate the final prediction.

The overall framework is illustrated in Figure 1. RAE-XMC decomposes $p(y_\ell \mid q)$, the conditional probability of label $\ell$ being relevant to a test input $q$, into two steps: (1) *retrieve-from-memory* and (2) *predict*. Given the test input $q$, we first retrieve possibly relevant keys $k$ from a knowledge memory $\mathcal{K}$, which can be modeled by a sampling process from the retriever distribution $p(k \mid q; \theta)$. Next, conditioned on both the retrieved key $k$ and the test input $q$, we define the predictive score for label $y_\ell$, denote as $p(y_\ell \mid k, q)$. In the second step, we compute the predictive score $p(y_\ell \mid k, q)$ for the

label $y_\ell$. Specifically, the inference process of RAE-XMC is defined as

$$p(y_\ell \mid q) = \sum_{k \in \mathcal{K}} \underbrace{p(y_\ell \mid k, q)}_{\text{Predictor}} \cdot \underbrace{p(k \mid q; \theta)}_{\text{Retriever}}. \qquad (2)$$

In Section 3.2, we further reveal the modeling choice of the parametrized knowledge retriever $p(k \mid q; \theta)$ and the lightweight (non-parametric) knowledge-augmented predictor $p(y_\ell \mid k, q)$ .

## 3.2 MODEL ARCHITECTURE

**Knowledge Memory.**   For XMC tasks, we define the knowledge memory $\mathcal{K}$ as the union of the input instance space $\mathcal{X}$ and the label space $\mathcal{L}$, namely $\mathcal{K} = \mathcal{X} \cup \mathcal{L}$. In other words, the knowledge memory $\mathcal{K}$ can be viewed as a set of key value pairs, where the key is input/label embeddings and the value is the corresponding one-hot label vectors. We now present the knowledge memory in the matrix form $\mathcal{K} = (\mathbf{K}, \mathbf{V})$, as defined as

$$\mathbf{K} = [\mathbf{x}_1^\top, \ldots, \mathbf{x}_N^\top, \mathbf{z}_1^\top, \ldots, \mathbf{z}_L^\top] = [\mathbf{X}, \mathbf{Z}] \in \mathbb{R}^{(N+L) \times d} \qquad (3)$$

$$\mathbf{V} = [\lambda \mathbf{Y}, (1-\lambda)\mathbf{I}_L] \in [0, 1]^{(N+L) \times L}, \qquad (4)$$

where the key matrix $\mathbf{K}$ consists of row-wise stack of training input embeddings $\mathbf{x}_i, \forall i = 1, \ldots N$, followed by label embeddings $\mathbf{z}_\ell, \forall \ell = 1, \ldots, L$. The value matrix $\mathbf{V}$ consists of row-wise stack of the ground truth label matrix $\mathbf{Y}$ from training set and the diagonal one matrix $\mathbf{I}_L$. Note that we introduce a coefficient $\lambda$ in $\mathbf{V}$, which trade-off the impact of predictive scores between the input space $\mathcal{X}$ and the label space $\mathcal{L}$. We further discuss the impact of $\lambda$ on inference in Section 3.3, as well as its performance in Section 5.3 and Appendix B.2.

**Knowledge Retriever.**   Given the test input $q$, the retriever of RAE-XMC defines the Softmax distribution over all relevance scores in $\mathcal{K}$:

$$p(k \mid q; \theta) = \frac{\exp\left(s_\theta(q, k)/\tau\right)}{\sum_{k' \in \mathcal{K}} \exp\left(s_\theta(q, k')/\tau\right)} \sim \text{Softmax}(\mathbf{q}^\top \mathbf{K}^\top / \tau), \qquad (5)$$

where the underlying scorer is a dense embedding-based model $s_\theta(k, q) = \langle f_\theta(k), f_\theta(q) \rangle$ and $\tau$ is the temperature controlling the skewness of the Softmax distribution. Note that we adopt the shared encoder setup for both input $q$ and label $\ell$, which is commonly-used in the XMC literature (Dahiya et al., 2023a;b; Gupta et al., 2024). For the embedding function, we consider average pooling of the hidden states from the last layer of the BERT-based Transformer encoder and apply L2-normalization to project the $d$-dimensional vector into the unit-sphere $\mathbb{S}^{d-1}$: $\mathbf{k} = f_\theta(k)/\|f_\theta(k)\|_2$ and $\mathbf{q} = f_\theta(q)/\|f_\theta(q)\|_2$.

**Lightweight Predictor.**   Given the test input $q$ and a retrieved key $k$, the knowledge-augmented predictor defines predictive score for label $\ell$ via $p(y_\ell \mid k, q)$. For knowledge-intense NLP applications that *do not* have strict real-time latency constraints, it is common to learn $p(y_\ell \mid k, q)$ via another complex language model (parameterized by $\psi$), which can be encoder-only LMs (Guu et al., 2020), encoder-decoder LMs (Lewis et al., 2020; Izacard et al., 2023) or even decoder-only LMs (Borgeaud et al., 2022). While large LMs excel at processing retrieved documents (Gao et al., 2024), achieving optimal performance on domain-specific XMC datasets may necessitate additional fine-tuning, rendering it impractical to scale to extremely enormous label spaces.

On the other hand, practitioners in the XMC community (Etter et al., 2022; Yu et al., 2022) often focus on real-time retrieving labels from the extremely large output space, which has high requirements on the inference latency. Thus, for the XMC tasks, we aim to design a lightweight predictor by directly looking up the $k$th row and $\ell$ columns of the ground truth Value matrix $\mathbf{V}_{k,\ell}$ given the retrieved $k$ and the desirable label $\ell$:

$$p(y_\ell \mid k, q) = \mathbf{V}_{k,\ell} \in \{0, 1\}. \qquad (6)$$

Note that the retrieved key $k \in \mathcal{X} \cup \mathcal{L}$ may come from either the training input space $\mathcal{X}$ or the label space $\mathcal{L}$. When $k \in \mathcal{X}$, we are then essentially using the ground truth label matrix $\mathbf{Y}_{k,\ell}$ as the predictions (cf., Eq 4). When $k \in \mathcal{L}$, $V_{k,\ell} = 1$ iff $k = \ell$ due to the diagonal matrix $\mathbf{I}_L$ in Equation 4, which falls back to using the retrieved label $\ell$ from the knowledge retriever as the predictions.

## 3.3 Inference of RAE-XMC

With the modeling choice of knowledge retriever $p(k \mid q; \theta)$ and lightweight predictor $p(y_\ell \mid k, q)$ in Section 3.2, we can rewrite Equation 2 into the matrix form as:

$$\hat{\mathbf{p}} = \mathrm{Softmax}(\mathbf{q}^\top \mathbf{K}^\top / \tau) \mathbf{V} \in \mathbb{R}^{1 \times L}, \tag{7}$$

which outputs $L$-dimensional predictive scores $\hat{\mathbf{p}} = [p(y_1 \mid q), \ldots, p(y_\ell \mid q), \ldots, p(y_L \mid q)] \in \mathbb{R}^L$.

Note that $\lambda$ in $\mathbf{V}$ (c.f., Eq 4) plays a crucial role. For $\lambda = \{0.0, 1.0\}$, the inference becomes

$$\hat{\mathbf{p}} = \begin{cases} \mathrm{Softmax}(\mathbf{q}^\top \mathbf{Z}^\top / \tau), & \text{if } \lambda = 0.0, \\ \mathrm{Softmax}(\mathbf{q}^\top \mathbf{X}^\top / \tau) \mathbf{Y}, & \text{if } \lambda = 1.0. \end{cases} \tag{8}$$

From the above, we see that the inference of RAE-XMC with $\lambda = 0.0$ is equivalent to the inference procedure of dual-encoder models, where the relevance score $\hat{\mathbf{p}}_\ell$ is solely determined by the test input embedding $\mathbf{q}$ and label embedding $\mathbf{z}_\ell$. On the other hand, the inference of RAE-XMC with $\lambda = 1.0$ resembles the classical non-parametric kNN classifiers for multi-label learning (Zhang & Zhou, 2007), where the relevance score $\hat{\mathbf{p}}_\ell$ is determined by the votes from the labels of retrieved training instances.

**Implementation.** Computing relevance scores for all keys in the Softmax distribution costs $\mathcal{O}(N + L)$, which is prohibitively expensive for XMC problems. Instead, we consider taking only top $b$ keys with highest probability under $\mathrm{Softmax}(\mathbf{q}^\top \mathbf{K}^\top / \tau)$, which is reasonable if most keys have near zero probability under certain temperatures. In Appendix B.1, we found the performance consistently increases with larger $b$, and saturates after some point that is sufficient to approximate the full Softmax distribution. We employ Approximate Nearest Neighbor Search (ANN) algorithms to find top $b$ keys efficiently. For example, the graph-based ANN method (e.g., HNSWLib (Malkov & Yashunin, 2018)) has a search complexity of $\mathcal{O}((\log(N + L))$. The pseudo code of the inference procedure is shown in Algorithm 1, and the indexing algorithm (that built the indexer and $\mathbf{V}$) can be found at Appendix A.3.

**Time Complexity.** For real-time inference (i.e., batch size of $1$), the time complexity of Algorithm 1 is $\mathcal{O}\big(\mathcal{C}(f_\theta) + \log(N + L) + b \log(L)\big)$. $\mathcal{O}(\mathcal{C}(f_\theta))$ is the complexity of embedding the test input $\mathbf{q}$. $\mathcal{O}(\log(N + L))$ is the ANN search time complexity. $\mathcal{O}(b \log(L))$ is the complexity of sparse matrix vector multiplication between $\mathbf{q}^\top \mathbf{K}^\top$ and $\mathbf{V}$, where we assume the average number of positive labels per input follows $\bar{L} = \log(L)$, which is a commonly-used assumption in the XMC literature (Yen et al., 2016; Prabhu et al., 2018).

---

**Algorithm 1** Inference of RAE-XMC

```
def InferenceRAE(Q_txt, f_enc, indexer, V_mat, b, tau=0.04):
    Q_emb = f_enc(Q_txt) # [bsz, d]
    QKT = indexer.search(Q_emb, topk=b) # [bsz, (N+L)], a sparse matrix
    QKT = Softmax(-QKT / tau, axis=1)
    return QKT.dot(V_mat) # [bsz, L] = [bsz, (N+L)] * [(N+L), L]
```

---

## 3.4 Training of RAE-XMC

Our proposed method differs from conventional approaches, which require intensive encoder training to capture subtle differences in label descriptions and enhance memorization. In contrast, a standard InfoNCE contrastive loss in Equation 1, is sufficient to achieve strong performance within the RAE-XMC inference framework.

To pursue optimal performance, as noted by DEXML (Gupta et al., 2024), using other positive labels as negatives for the current positive label leads the loss function to penalize positive labels that are easier to predict. We further apply decoupled softmax loss, which removes the positive labels $P(\mathbf{y}) = \{\ell \mid \forall \ell \in [L] : y_\ell = 1\}$ from the denominator:

$$J(x, \mathbf{y}; s_\theta) = -\sum_{\ell \in [L]} y_\ell \cdot \log \frac{e^{s_\theta(x, \ell)/\tau}}{e^{s_\theta(x, \ell)/\tau} + \sum_{\forall \ell' \notin P(\mathbf{y})} e^{s_\theta(x, \ell')/\tau}}. \tag{9}$$

RAE-XMC performs inference over a joint input instance and label space. To enhance consistency between inference and training, we propose utilizing in-batch input instances as negatives (Moiseev et al., 2023). This approach forces the model to contrast both instance and label spaces to accurately classify the correct label. The training objective of RAE-XMC is:

$$J(x, \mathbf{y}; s_\theta) = -\sum_{\ell \in L} y_\ell \cdot \log \frac{e^{s_\theta(x,\ell)/\tau}}{e^{s_\theta(x,\ell)/\tau} + \sum_{\forall \ell' \notin P(\mathbf{y})} e^{s_\theta(x,\ell')/\tau} + \sum_{x' \in Q(\mathbf{y})} e^{s_\theta(x,x')/\tau}}, \quad (10)$$

where $Q(\mathbf{y}) = \{x' \mid \forall (x', \mathbf{y}') \in B : P(\mathbf{y}) \cap P(\mathbf{y}') = \emptyset\}$ and $B$ is the mini-batch of training set $\mathcal{D}$. $Q(\mathbf{y})$ is the set of the negative input instance $x'$ that do not share any positive labels with the current input $x$, which prevents pushing of input instances with shared labels further apart.

**Implementation.** In practice, it is infeasible to sum over all the labels to accurately estimate the model predicted label distribution without using gradient caching (Guo et al., 2020). To approximate the label distribution, for each input $x$, we randomly sample one positive label from $P(\mathbf{y})$ and sample $m$ negative labels through hard negative mining.

## 4 RELATED WORK

**Extreme Multi-label Classification.** Conventional XMC methods finetune Transformer encoders and learn one-versus-all (OVA) label classifiers for the XMC tasks (Chang et al., 2020b; Zhang et al., 2021; Kharbanda et al., 2022). To tackle the large output space challenge, various label space partitioning techniques or surrogate loss functions have been extensively studied (Prabhu et al., 2018; Jain et al., 2019; Yu et al., 2022). However, these approaches treat labels as featureless identifiers and learn classification vector for each label, which may not generalize to unseen labels. On the other hand, there are increasing number of work leverage the label text feature by employing Dual-Encoder (DE) models that learn embeddings from both input and label text (Saini et al., 2021; Mittal et al., 2021; Dahiya et al., 2021; 2023a;b). However, the two representative methods in this line of work, namely NGAME (Dahiya et al., 2023a) and DEXA (Dahiya et al., 2023b), do not solely rely on a DE model. Specifically, NGAME is a two-stage approach that involves first training input and label embeddings via a DE model and then utilizing a classification network in the second stage. DEXA builds on NGAME to improve the encoder embeddings by augmenting them with auxiliary parameters. Very recently, DEXML (Gupta et al., 2024) is a pure DE model (without learning any label classifier) that achieves SOTA results by using a decoupled InfoNCE loss function. Nevertheless, DEXML suffers from extremely long training time issue as it relies on gradient cache (Gao et al., 2021) to have a precise estimation of the label Softmax distribution that costs $\mathcal{O}(L)$ per training step.

**Retrieval Augmented Language Models.** Using external knowledge to improve deep neural networks has been widely explored in the context of retrieval-augmented language models (Khandelwal et al., 2020; Guu et al., 2020; Lewis et al., 2020; Izacard et al., 2023; Borgeaud et al., 2022; Shi et al., 2023). For example, kNN-LM (Khandelwal et al., 2020) interpolates the next-token probability by the neural language model and the kNN model at inference stage. REALM (Guu et al., 2020) and many of its follow-up work (Izacard et al., 2023; Borgeaud et al., 2022) consider learning the knowledge retriever jointly with the underlying language models. While sharing similar intuition to the proposed RAE-XMC framework, both kNN-LM and REALM are designed for language modeling tasks, which did not consider a multi-label formulation of the knowledge source.

**Retrieval Augmented Multi-label Text Classification.** Retrieval-augment techniques have also been applied to *small-scale* multi-label text classifications (Chalkidis & Kementchedjhieva, 2023; Wang et al., 2022), where the number of labels is only a few hundreds. Specifically, Chalkidis & Kementchedjhieva (2023) considers a multi-stage ranking setup: the first stage learns a text encoder and OVA label classifiers to build the knowledge memory; the second stage trains an expensive cross-attention encoder to produce scores given the test input and the retrieved top-K labels from the first stage. Wang et al. (2022) considers training a text encoder and OVA label classifiers with the contrastive loss. At inference time, they interpolates the OVA label classifier scores with the top-K instances' label one-hot vector retrieved from the instance space.

We discussed some major difference between Wang et al. (2022) and our RAE-XMC framework: Regarding problem setups, they focus on small-scale multi-label problems with hundreds of labels and treat labels as featureless IDs. In contrast, RAE-XMC tackles extreme-scale multi-label (XMC) problems with millions of labels and leverage the label text information. Speaking of model architectures, they consider OVA label classifiers where the number of trainable parameters scales linearly to $L$, which is not scalable for XMC tasks. On the other hand, RAE-XMC is parametrized by Dual-Encoders where the number of trainable parameters does not scale with the number of labels $L$. Finally, for prediction functions, they conduct two ANN searches: one from the kNN classifier on the input space and the other from the label OvA classifier on the label space, to form final prediction scores. On the contrary, RAE-XMC conducts a single ANN search on the union of input and label space. See empirical comparison in Table 1 and Table 4 for more details.

## 5 EXPERIMENTS

**Datasets and Evaluation.** We conduct experiments on four LF-XMC datasets, including LF-AmazonTitles-131K, LF-WikiSeeAlso-320K, LF-Wikipedia-500K, and LF-AmazonTitles-1.3M, where the prefix 'LF' denotes the datasets contain label descriptions. Details of these datasets and its statistics are presented in Table 6 of Appendix A. We use the same raw text input and training/test data splits as in (Gupta et al., 2024) to have a fair and reproducible comparison. Following the evaluation setup from XMC Repository (Bhatia et al., 2016) and XMC literature (Yu et al., 2022; Gupta et al., 2024; Schultheis et al., 2024b), we consider Precision (P@1 and P@5), Recall (R@100), and Macro average F1 to measure the standard XMC and retrieval metrics.

**Baselines.** We compare RAE-XMC to competitive XMC baselines, which fall into two categories. The first category is dual-encoder (DE) models without learning OVA label classifiers, including DPR (Karpukhin et al., 2020), ANCE (Xiong et al., 2021), NGAME$_{DE}$ (Dahiya et al., 2023a), DEXA$_{DE}$ (Dahiya et al., 2023b) and DEXML (Gupta et al., 2024); The second category is deep learning models with learning OVA label classifiers, such as XR-Transformer (Zhang et al., 2021) NGAME$_{DE+OVA}$ (Dahiya et al., 2023a), DEXA$_{DE+OVA}$ (Dahiya et al., 2023b), and OvA+kNN (Wang et al., 2022). For OvA Classifier, we apply One-versus-All Binary Cross-Entropy (OvA-BCE) as the training loss, and OvA+kNN Wang et al. (2022) applies kNN prediction on top of the OvA Classifier model, as described in Equation 11.

RAE-XMC falls into the first category, which has much smaller model parameters to learn. Thus, we consider the state-of-the-art (SOTA) DE method, DEXML as our main baseline. To have a fair comparison, we use the same 66M parameter distilbert-base transformer encoder (Sanh et al., 2020) for RAE-XMC, as used in NGAME$_{DE}$, DEXA$_{DE}$, DEXML, OvA Classifier and OvA+kNN (Wang et al., 2022). All the experiments are conducted on an AWS p4d.24xlarge instance, with 8 Nvidia A100 GPUs and 96 CPUs. See Appendix A for more details on the experiment setup and hyperparameters of RAE-XMC. Our experiment code is available in the Supplementary Material.

### 5.1 MAIN RESULTS

Table1 presents the main results and the training time (in hours). For large-scale datasets, such as LF-Wikipedia-500K and LF-AmazonTitles-1.3M, RAE-XMC not only advances the current SOTA results of DEXML (Gupta et al., 2024), but also achieves significant speedup in training time. Specifically, RAE-XMC has a speedup of 6x and 18x over DEXML, for LF-Wikipedia-500K and LF-AmazonTitles-1.3M, respectively. Compared with OvA+kNN (Wang et al., 2022) which also leverages retrieval-augmentation, RAE-XMC achieves significantly better performance, attributed to a more effective training loss and an improved prediction function. Compared with the complex two-stage XMC methods that learn both a DE model and trainable OVA label classifiers, RAE-XMC outperforms NGAME$_{DE+OVA}$ and DEXA$_{DE+OVA}$ across all four datasets, except P@1 on the smallest LF-AmazonTitles-131K. Note that the number of trainable parameters of RAE-XMC is significantly smaller than OvA+kNN, NGAME$_{DE+OVA}$ and DEXA$_{DE+OVA}$, which scales linearly to the size of label space. As the label space increases, the memory and training difficulty for OVA classifiers increase accordingly, which makes it challenging to apply to large-scale datasets.

Table 1: Comparing RAE-XMC with recent XMC methods on four public LF-XMC datasets. Superscripts $\dagger$ and $\star$ indicate results excerpted from XMC Repository (Bhatia et al., 2016) and DEXA$_{DE}$ (Dahiya et al., 2023b), respectively. Superscripts $\ddagger$ denotes that we reproduce DEXML results by evaluating their official released model checkpoints. TT denotes training time in hours. For DEXML and RAE-XMC, TT is measured on 8 A100 GPUs. Blank entries indicate source does not have those numbers. We conduct significant test between RAE-XMC and DEXML on three metrics (P@1, P@5, R@100) across all datasets. All results are significant (p-values smaller than $10^{-10}$) except for P@1 on LF-AmazonTitles-1.3M. See Appendix B.3 for more details.

| Methods | OvA Label Classifier | P@1 | P@5 | R@100 | TT | P@1 | P@5 | R@100 | TT |
|---|---|---|---|---|---|---|---|---|---|
| | | LF-AmazonTitles-131K | | | | LF-WikiSeeAlso-320K | | | |
| OvA Classifier | ✓ | 37.13 | 18.32 | 64.93 | 1.2 | 41.10 | 20.43 | 72.81 | 2.2 |
| OvA+kNN | ✓ | 38.87 | 19.19 | 65.96 | 1.2 | 43.30 | 21.91 | 73.39 | 2.2 |
| XR-Transformer$^\dagger$ | ✓ | 38.10 | 18.32 | - | 35.4 | 42.57 | 21.30 | - | 119.5 |
| NGAME$^\dagger_{DE+OVA}$ | ✓ | 46.01 | 21.47 | - | 12.6 | 47.65 | 23.68 | - | 75.4 |
| DEXA$^\dagger_{DE+OVA}$ | ✓ | **46.42** | 21.59 | - | 13.0 | 47.11 | 22.71 | - | 78.6 |
| DPR$^\star$ | ✗ | 41.85 | 20.88 | - | - | 41.66 | 20.66 | - | - |
| ANCE$^\star$ | ✗ | 42.67 | 20.98 | - | - | 44.35 | 21.99 | - | - |
| NGAME$^\star_{DE}$ | ✗ | 42.61 | 20.69 | - | - | 43.58 | 20.86 | - | - |
| DEXA$^\star_{DE}$ | ✗ | 44.76 | 21.18 | - | - | 46.57 | 22.26 | - | - |
| DEXML$^\ddagger$ | ✗ | 42.24 | 20.47 | 68.81 | 2.1 | 45.76 | 21.75 | 72.87 | 12.8 |
| RAE-XMC (ours) | ✗ | 45.10 | **21.95** | **71.94** | **0.6** | **48.04** | **23.68** | **79.92** | **0.6** |
| | | LF-Wikipedia-500K | | | | LF-AmazonTitles-1.3M | | | |
| OvA Classifier$^\star$ | ✓ | 82.00 | 48.54 | - | - | 48.72 | 39.09 | - | - |
| XR-Transformer$^\dagger$ | ✓ | 81.62 | 47.85 | - | 318.9 | 50.98 | 40.05 | - | 132.0 |
| NGAME$^\dagger_{DE+OVA}$ | ✓ | 84.01 | 49.97 | - | 54.9 | 56.75 | 44.09 | - | 97.8 |
| DEXA$^\dagger_{DE+OVA}$ | ✓ | 84.92 | 50.51 | - | 57.5 | 56.63 | 43.90 | - | 103.1 |
| DPR$^\star$ | ✗ | 65.23 | 35.23 | - | 54.7 | 44.64 | 34.83 | - | 96.8 |
| ANCE$^\star$ | ✗ | 63.33 | 33.12 | - | 75.1 | 46.44 | 37.59 | - | 447.3 |
| NGAME$^\star_{DE}$ | ✗ | 77.92 | 40.95 | - | 41.9 | 45.82 | 35.48 | - | 75.5 |
| DEXA$^\star_{DE}$ | ✗ | 79.99 | 42.52 | - | 42.8 | 51.92 | 38.86 | - | 76.6 |
| DEXML$^\ddagger$ | ✗ | 85.59 | 50.39 | **90.52** | 37.0 | 58.43 | 45.48 | 64.26 | 132.0 |
| RAE-XMC (ours) | ✗ | **86.49** | **50.67** | 90.24 | **6.4** | **58.48** | **47.00** | **66.88** | **7.4** |

## 5.2 PERFORMANCE VERSUS TRAINING EFFICIENCY

In Figure 2, we study the trade-off between model predictive power (i.e., Precision@1) versus the corresponding model training time. To match DEXML performance (Gupta et al., 2024), we see that RAE-XMC is extremely efficient to train. RAE-XMC achieves 154x, 20x and 18x speedup to achieve on par performance of DEXML, for LF-WikiSeeAlso-320K, LF-Wikipedia-500K, and LF-AmazonTitles-1.3M, respectively. As discussed in Section 3.3, the inference of RAE-XMC is equivalent to the inference of DE models when $\lambda = 0.0$, which is the green curve in Figure 2. Eventually, we expect RAE-XMC with $\lambda = 0.0$ progresses toward the performance of DEXML (the magenta line), while it may take significantly longer time because of the large-scale label space. On the other hand, RAE-XMC with $\lambda = 1.0$, the blue curve in Figure 2, reduces to the vanilla $k$NN classifier, which have strong predictive power at the early stage of training while saturating at the later stage. Finally, the proposed RAE-XMC method (w/ $\lambda = 0.5$) effectively combines the best of both world between the vanilla kNN classifier and the DE model, as shown in the red curve in Figure 2.

## 5.3 MEMORIZATION AND GENERALIZATION CAPABILITY

Similar to NGAME (Dahiya et al., 2023a) and DEXML (Gupta et al., 2024), we dissect the label space into four segments: head, torso, tail, and extreme tail (xTail), based on the label frequency. This study investigates the model's capability on memorization and generalization, where the model should excel at the head and xTail segments, respectively. In Table 2, we report Macro-average F1@5 metric on two largest LF-XMC datasets, which better measures the tail label performance (Zhang et al., 2023; Schultheis et al., 2024b;a).

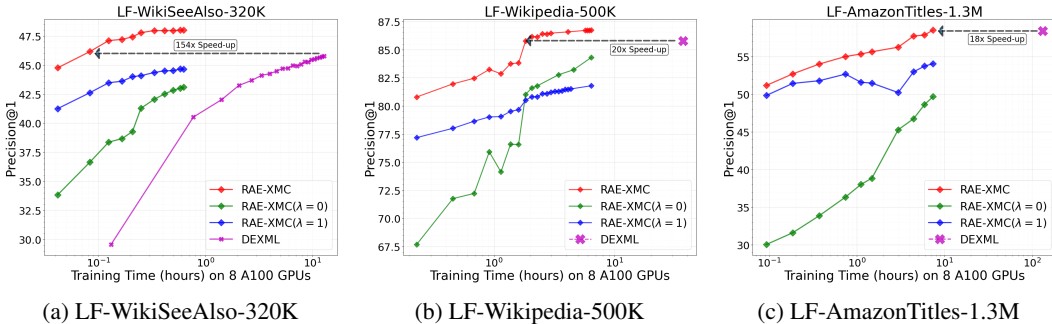

(a) LF-WikiSeeAlso-320K  (b) LF-Wikipedia-500K  (c) LF-AmazonTitles-1.3M

Figure 2: Model performance versus model training time on two large-scale LF-XMC datasets. Y-axis and X-axis are Precision@1 metric and training time in hours measured on 8 Nvidia A100 GPUs.

By varying $\lambda$ at inference stage, RAE-XMC demonstrates controllable trade-off between memorization and generalization, namely the performance on head and tail label segments, respectively. Take LF-AmazonTitles-1.3M as an example. Comparing RAE-XMC$_{\lambda \to 1}$ and RAE-XMC$_{\lambda \to 0}$, the former solely relies on the memorization while the latter solely relies on the generalization component, respectively. Thus, the former ($\lambda = 1$) has a higher F1@5 for the head label segment (i.e., 34.5 vs 15.9), while the latter ($\lambda = 0$) has a higher F1@5 for the xTail label segment (i.e., 12.0 vs. 22.0). With the selected $\lambda$, RAE-XMC equips the underling DE model with better memorization capability, which advances the current SOTA result of DEXML (Gupta et al., 2024) on the head label segment (i.e., 33.8 vs. 28.7).

Table 2: Macro-average F1@5 metrics under four label segments: Head, Torso, Tail, and Extreme Tail (xTail), which is based on label frequency of $\{(1000, N_{trn}), (100, 1000], (10, 100], (0, 10]\}$.

| Methods | Head | Torso | Tail | xTail |
|---|---|---|---|---|
| | LF-Wikipedia-500K | | | |
| DEXML (Gupta et al., 2024) | 72.1 | 43.6 | 40.1 | 41.1 |
| RAE-XMC | 73.8 | 43.0 | 39.5 | 39.1 |
| RAE-XMC$_{\lambda \to 0.0}$ | 60.8 | 49.4 | 44.7 | 43.1 |
| RAE-XMC$_{\lambda \to 1.0}$ | 71.1 | 36.7 | 29.6 | 21.6 |
| | LF-AmazonTitles-1.3M | | | |
| DEXML (Gupta et al., 2024) | 28.7 | 15.2 | 13.3 | 20.6 |
| RAE-XMC | 33.8 | 15.8 | 12.0 | 17.4 |
| RAE-XMC$_{\lambda \to 0.0}$ | 15.9 | 13.4 | 14.2 | 22.0 |
| RAE-XMC$_{\lambda \to 1.0}$ | 34.5 | 14.8 | 9.9 | 12.0 |

## 5.4 ABLATION ON TRAINING LOSS FUNCTIONS

As shown in Table 3, using in-batch input instances as negatives in Equation 9 consistently improves performance, except for P@1 on LF-Wikipedia-500K. This demonstrates the effectiveness of training the model to contrast over a joint label and instance embedding space. We found that hard-negative mining has a varied impact on different metrics. It consistently improves P@1, but sometimes negatively affects R@100. This is because hard-negative mining makes the model more selective, strictly removing labels that may have less relevance.

Table 3: Ablation studies of the training objective of RAE-XMC in Equation 9. "In-batch neg X" denotes whether we include negative inputs within the batch in the denominator of Equation 9. "Hard-Neg" denotes whether we conduct hard negative mining from the label space.

| Variants of RAE-XMC | | LF-WikiSeeAlso-320K | | | LF-Wikipedia-500K | | | LF-AmazonTitles-1.3M | | |
|---|---|---|---|---|---|---|---|---|---|---|
| In-batch neg X | Hard-Neg | P@1 | P@5 | R@100 | P@1 | P@5 | R@100 | P@1 | P@5 | R@100 |
| ✗ | ✓ | 46.51 | 22.97 | 78.79 | 86.71 | 49.75 | 89.18 | 58.12 | 46.75 | 66.46 |
| ✓ | ✗ | 47.84 | 23.72 | 80.50 | 84.25 | 48.55 | 91.43 | 57.94 | 46.82 | 67.22 |
| ✓ | ✓ | 48.04 | 23.68 | 79.92 | 86.49 | 50.67 | 90.24 | 58.43 | 47.00 | 66.88 |

## 5.5 ABLATION ON PREDICTION FUNCTIONS

In Equation 7, we retrieve keys from a unified instance and label memory $\mathbf{K}$ as part of our prediction function. Using the same encoder trained by RAE-XMC framework, we compare our inference method against the prediction function of OvA+kNN (Wang et al., 2022), which separately retrieves keys from input and label spaces, and performs the convex combination as final prediction scores:

$$\hat{\mathbf{p}} = \lambda * \text{Softmax}(\mathbf{q}^\top \mathbf{X}^\top / \tau)\mathbf{Y} + (1 - \lambda) * \text{Softmax}(\mathbf{q}^\top \mathbf{Z}^\top / \tau). \qquad (11)$$

As shown in Table 4, although both methods show improvement over the baselines (i.e., $\lambda = 0$ or $\lambda = 1$), the improvement of RAE-XMC is much greater. This shows the effectiveness of ranking keys in a unified memory, leading to better calibration when combining two different spaces.

## 5.6 RAE INFERENCE ON VARIOUS ENCODERS

In Table 5, we evaluate RAE inference on various encoders, including those with and without supervised fine-tuning (SFT). For the methods without RAE inference, we employ standard dual-encoder inference on these encoders. Comparing DEXML with DEXML +RAE, we consistently observe performance enhancements with our proposed inference method. However, the improvement is relatively modest, mainly because DEXML has already been well-trained and possesses strong memorization capability.

For the encoders without SFT, we observe significant performance gains with our proposed RAE inference. Without SFT, pre-trained encoders may struggle to align input instances and label descriptions in embedding space. Nonetheless, they can still retrieve relevant input instances to enhance the prediction. Surprisingly, on LF-AmazonTitles-1.3M, even without any fine-tuning, the performance of GTE$_{\text{base}}$ +RAE can achieve results on par with previous supervised DE methods, such as ANCE (Xiong et al., 2021) and NGAME$_{\text{DE}}$ (Dahiya et al., 2023a) [1]. This further underscores the effectiveness of our approach in significantly reducing the need for intensive training of the DE model.

Table 4: Ablation study of the prediction function of RAE-XMC in Eq 7 on LF-AmazonTitles-1.3M.

| $\lambda$ | OvA+kNN prediction | | | RAE-XMC prediction | | |
|---|---|---|---|---|---|---|
| | P@1 | P@5 | R@100 | P@1 | P@5 | R@100 |
| 0 | 49.72 | 39.45 | 61.18 | 49.72 | 39.45 | 61.18 |
| 0.30 | 53.59 | 44.56 | 64.14 | 57.37 | 47.03 | 67.16 |
| 0.50 | **53.95** | **44.10** | **63.52** | **58.48** | **47.00** | **66.88** |
| 0.70 | 55.37 | 43.96 | 62.95 | 58.60 | 46.81 | 66.65 |
| 1.00 | 53.98 | 43.25 | 61.67 | 53.98 | 43.25 | 61.67 |

Table 5: RAE inference results on various Encoders. SFT denotes "supervised fine-tuning".

| Encoders | SFT | P@1 | P@5 | R@100 |
|---|---|---|---|---|
| | | LF-WikiSeeAlso-320K | | |
| DEXML (Gupta et al., 2024) | ✓ | 45.77 | 21.75 | 72.87 |
| +RAE (ours) | ✓ | 46.23 | 22.06 | 72.23 |
| DistilBERT (Sanh et al., 2020) | ✗ | 2.47 | 1.36 | 14.19 |
| +RAE (ours) | ✗ | 16.82 | 9.38 | 56.99 |
| GTE$_{\text{base}}$ (Li et al., 2023) | ✗ | 22.53 | 11.57 | 54.83 |
| +RAE (ours) | ✗ | 36.52 | 19.32 | 72.84 |
| | | LF-AmazonTitles-1.3M | | |
| DEXML (Gupta et al., 2024) | ✓ | 58.40 | 45.46 | 64.25 |
| +RAE (ours) | ✓ | 58.83 | 45.76 | 65.55 |
| DistilBERT (Sanh et al., 2020) | ✗ | 21.40 | 11.86 | 17.16 |
| +RAE (ours) | ✗ | 35.59 | 28.65 | 46.44 |
| GTE$_{\text{base}}$ (Li et al., 2023) | ✗ | 27.16 | 16.56 | 27.50 |
| +RAE (ours) | ✗ | 47.03 | 37.73 | 56.01 |

## 6 CONCLUSION AND LIMITATIONS

We introduce the novel RAE-XMC framework for XMC problems, which enhances the DE model with retrieval-augmented capability and achieves new SOTA results for large-scale LF-XMC datasets. While being effective and highly efficient to train, RAE-XMC also comes with two limitations. First, the knowledge memory will bring additional inference overhead, both in space and time complexity. How to select representative key-value pairs to reduce the size of knowledge memory is an interesting and challenging avenue for future work. In addition, the performance of RAE-XMC may be hindered if there is huge discrepancy between the knowledge memory (training corpus and label space) versus the test instances. How to mitigate this issue by learning data-driven $\lambda$ in the value matrix is another interesting direction, which we save for future work.

---

[1] We use gte-base-en-v1.5 in `https://huggingface.co/spaces/mteb/leaderboard` as our GTE$_{\text{base}}$ model. Note that the performance may be further improved with more advanced encoders.

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

# A IMPLEMENTATION DETAILS

## A.1 DATASETS

Following standard XMC literature, we downloaded the datasets from a public XMC repository[2], which already splits the data into training and testing sets.

Table 6: Data Statistics. $N_{\text{train}}$ and $N_{\text{test}}$ refer to the number of instances in the training and test set, respectively. $L$: the number of labels. $\bar{L}$: the average number of positive labels per instance. $\bar{N}$: the average number of instances per label.

| Dataset | $N_{\text{train}}$ | $N_{\text{test}}$ | $L$ | $\bar{L}$ | $\bar{N}$ |
|---|---|---|---|---|---|
| LF-AmazonTitles-131K | 294,805 | 134,835 | 131,073 | 2.29 | 5.15 |
| LF-WikiSeeAlso-320K | 693,082 | 177,515 | 312,330 | 2.11 | 4.67 |
| LF-Wikipedia-500K | 1,813,391 | 783,743 | 501,070 | 4.74 | 17.15 |
| LF-AmazonTitles-1.3M | 2,248,619 | 970,237 | 1,305,265 | 22.20 | 38.24 |

## A.2 EVALUATION METRICS

For P@k (precision at k), we report $\frac{\text{TP@k}}{\text{k}}$, where TP@k denotes the number of true positives in the top k ranking list. For R@k, we report $\frac{\text{TP@k}}{\text{TP@k+FN@k}}$, where FN@k denotes false negatives in the top k ranking list.

To obtain macro-average F1@k, we average the each label F1 score over a label segment. For each label $l$, we compute F1$_l$@k as $2 \cdot \frac{P_l@\text{k} \cdot R_l@\text{k}}{P_l@\text{k} + R_l@\text{k}}$. The $P_l@\text{k} = \frac{\text{TP}_l@\text{k}}{\text{TP}_l@\text{k} + \text{FP}_l@\text{k}}$, and the $R_l@\text{k} = \frac{\text{TP}_l@\text{k}}{\text{TP}_l@\text{k} + \text{FN}_l@\text{k}}$. The TP$_l$@k, FP$_l$@k, and FN$_l$@k denote true positives, false positives, and false negatives respectively for each label $l$ in the top k ranking list.

## A.3 HYPER-PARAMETERS OF RAE-XMC

All experiments take place on an AWS p4d.24xlarge instance, which has 8 Nvidia A100 GPUs (40 GB memory per GPU) and 96 Intel CPUs. The CPU memory is 1.10TB in total.

**Reproducibility** Our PyTorch based implementation is available as a .zip file in the Supplementary Material. We plan to open-source the code after the review period along with the trained models to facilitate further research in these directions. Below we provide additional details to clarify our approach and experimental setup.

**Training Hyper-parameters.** For fair comparison, we use the same 66M parameter distilbert-base transformer encoder (Sanh et al., 2020) as used in NGAME$_{\text{DE}}$ (Dahiya et al., 2023a), DEXA$_{\text{DE}}$ (Dahiya et al., 2023b), and DEXML (Gupta et al., 2024), unless otherwise specify. We use average pooling of the last hidden states of the Transformer encoder as the final input/label embeddings, followed by l2-normalization to project embeddings onto the unit-sphere. As the input/label embeddings are l2-normalized, the similarity scores are bounded between $[-1, 1]$, thus we fix the Softmax temperature $\tau$ to be $0.04$, which approximately amounts to $1/\sqrt{d}$ where $d$ is the embedding dimension. We train RAE-XMC with the AdamW optimizer and use linear decay with the warm-up learning rate schedule.

Rest of the training hyper-parameters considered in our experiments are described below

- max_len: maximum length of the input text to the transformer encoder. For input instance text length, we use 128 for LF-WikiSeeAlso-320K and 192 for LF-Wikipedia-500K. For short-text datasets (LF-AmazonTitles-131K and LF-AmazonTitles-1.3M), we set it to 32. For the label text, we always use a max length of 32.

---

[2]http://manikvarma.org/downloads/XC/XMLRepository.html

- bsz/GPU: the batch size per GPU for training. The global batch size is equal to $8 \times$ bsz/GPU.

- LR: the learning rate of the optimizer.

- max_steps: the maximum number of training steps in optimization.

- hnm_steps: the frequency to perform hard negative mining (HNM) during training.

- hnm_topk: the number of top-$k$ predicted labels used as the source of hard negatives.

- $m$: the number of hard negative labels sampled for each input $x$.

Table 7: Training Hyper-parameters of RAE-XMC

| Dataset | max_len | bsz/GPU | LR | max_steps | hnm_steps | hnm_topk | $m$ |
|---|---|---|---|---|---|---|---|
| LF-AmazonTitles-131K | 32 | 896 | 3e-4 | 3K | 1K | 50 | 2 |
| LF-WikiSeeAlso-320K | 128 | 576 | 2e-4 | 3K | 1K | 50 | 2 |
| LF-Wikipedia-500K | 192 | 400 | 2e-4 | 28K | 7K | 25 | 2 |
| LF-AmazonTitles-1.3M | 32 | 896 | 3e-4 | 40K | 8K | 50 | 2 |

**Inference Hyper-parameters.** We use HNSW algorithm (Malkov & Yashunin, 2018) to perform approximate nearest neighbor (ANN) search at the inference stage. The ANN index building procedure is described in Algorithm 2. To build the HNSW index, we set the maximum edge per node $M = 64$ and the queue size as $efC = 500$. The inference procedure of RAE-XMC is described in Algorithm 1. For training/inference consistency, we keep the temperature $\tau = 0.04$. We set the queue size $efS = 300$ for the HNSW searcher and select top $b = 200$ keys from the knowledge memory to approximate the Softmax distribution of the knowledge retriever. We set $\lambda = 0.5$ for all datasets except for LF-Wikipedia-500K, which has $\lambda = 0.01$. The effect of using different $\lambda$ is discussed in Appendix B.2.

---

**Algorithm 2** Indexing of RAE-XMC

---

```
def IndexingRAE(f_enc, X_txt, L_txt, Y_trn, lamb=0.5):
    X_emb = f_enc(X_txt) # [N, d]
    Z_emb = f_enc(L_txt) # [L, d]
    K_emb = vstack([X_emb, Z_emv]) # [(N+L), d], the Key matrix in Eq(3)
    ann_index = ANN.train(K_emb, ...) # Build ANN index on [N+L] space
    V_mat = vstack([lamb*Y_trn, (1-lamb)*I_L] # [(N+L), L], the Value matrix
    return (indexer, V_mat)
```

---

## B  ADDITIONAL EXPERIMENT RESULTS

### B.1  ANALYSIS OF SAMPLED KEY NUMBER

In Section 3.3, we retrieve top $b$ keys to approximate the softmax distribution. With greater $b$, the performance increases accordingly, but with higher inference latency as a tradeoff. In Figure 3, we examine the impact of different $b$ regarding the ratio of retrieved query number / retrieved label number and the performance.

Regarding the ratio of retrieved query number to retrieved label number, distinct trends emerge between LF-AmazonTitles-1.3M and LF-Wikipedia-500K datasets. In LF-AmazonTitles-1.3M, the ratio is lower, suggesting a higher reliance on label retrieval for predictions. Moreover, in LF-AmazonTitles-1.3M, the ratio tends to increase with larger values of $b$, whereas in LF-Wikipedia-500K, it initially rises and then declines. As $b$ increases, performance across all metrics consistently improves, but with varying impacts on individual metrics. Achieving optimal performance in @k metrics require higher values of $b$ as $k$ increases.

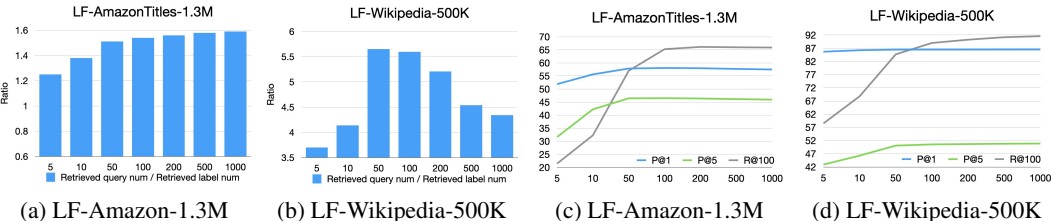

(a) LF-Amazon-1.3M     (b) LF-Wikipedia-500K     (c) LF-Amazon-1.3M     (d) LF-Wikipedia-500K

Figure 3: Analysis of different sources of top-$b$ keys retrieved from the knowledge retriever as well as the relation to the model performance.

Table 8: Comparison of different $\lambda$.

| Dataset | $\lambda = 0$ | | $\lambda = 0.01$ | | $\lambda = 0.1$ | | $\lambda = 0.3$ | | $\lambda = 0.5$ | | $\lambda = 0.7$ | | $\lambda = 0.9$ | | $\lambda = 1.0$ | |
|---|---|---|---|---|---|---|---|---|---|---|---|---|---|---|---|---|
| | P@1 | P@5 | P@1 | P@5 | P@1 | P@5 | P@1 | P@5 | P@1 | P@5 | P@1 | P@5 | P@1 | P@5 | P@1 | P@5 |
| LF-AmazonTitles-131K | 42.58 | 20.57 | 42.86 | 21.35 | 44.17 | 21.86 | 44.72 | 21.98 | 45.16 | 21.95 | 45.13 | 21.90 | 44.92 | 21.82 | 44.4 | 21.68 |
| LF-WikiSeeAlso-320K | 43.11 | 20.29 | 44.95 | 21.89 | 47.25 | 23.27 | 48.11 | 23.67 | 48.02 | 23.59 | 47.56 | 23.32 | 46.52 | 22.83 | 44.64 | 22.11 |
| LF-Wikipedia-500K | 84.30 | 50.39 | 86.49 | 50.67 | 86.71 | 48.15 | 86.11 | 46.19 | 85.37 | 45.12 | 84.49 | 44.29 | 83.25 | 43.50 | 81.79 | 42.90 |
| LF-AmazonTitles-1.3M | 49.41 | 38.71 | 49.61 | 41.90 | 54.36 | 45.87 | 57.24 | 46.55 | 57.98 | 46.39 | 57.89 | 46.15 | 57.43 | 45.83 | 51.93 | 41.61 |

## B.2 COMPARISON OF DIFFERENT KNOWLEDGE SOURCE FOR PREDICTION

In Table 8, we examine the performance variation of using different value of $\lambda$ to compare the impact of using various knowledge source for inference. By gradually increasing the value of $\lambda$, the weight of $k$NN classifier increases accordingly. Comparing $\lambda = 0.0$ and $\lambda = 0.01/0.1$, we can find that even though the prediction weight of the kNN classifier is very small, it can still complement the dual-encoder prediction very well, thus improving the performance.

Across all datasets excluding LF-Wikipedia-500K, RAE-XMC exhibits insensitivity to $\lambda$ variations. Notably, setting $\lambda$ to 0.5 suffices to attain optimal performance across most datasets, suggesting an equitable importance of both knowledge sources in prediction. However, when there is a distinct disparity between training and testing corpora, meticulously tuning $\lambda$ may be necessary (Grave et al., 2017; Khandelwal et al., 2020; Chang et al., 2024).

## B.3 SIGNIFICANT TEST

We conduct significant tests to verify the effectiveness of RAE-XMC in Table 1. In particular, we first compute the instance-wise metrics (i.e., P@1, P@5 and R@100) for the 1st and 2nd place method in Table 1, namely RAE-XMC and DEXML, respectively. Then we perform the paired t-test between RAE-XMC and DEXML for each of these metrics. As shown in Table 9, the results are significant on all the metrics, except P@1 of LF-AmazonTitles-1.3M.

Table 9: P-values of significant test between RAE-XMC and DEXML on instance-wise metrics.

| Dataset | P@1 | P@5 | R@100 |
|---|---|---|---|
| LF-AmazonTitles-131K | $9.5 \times 10^{-51}$ | $1.7 \times 10^{-67}$ | $4.9 \times 10^{-80}$ |
| LF-WikiSeeAlso-320K | $2.5 \times 10^{-42}$ | $2.3 \times 10^{-152}$ | $\approx 0.0$ |
| LF-Wikipedia-500K | $5.7 \times 10^{-59}$ | $1.1 \times 10^{-10}$ | $4.4 \times 10^{-23}$ |
| LF-AmazonTitles-1.3M | 0.51 | $2.24 \times 10^{-170}$ | $\approx 0.0$ |

