# OpenReview forum: "Retrieval-augmented Encoders for Extreme Multi-label Text Classification"
_ICLR.cc/2025/Conference — Submitted to ICLR 2025_

### Official Review · Reviewer_FvxB · 2024-10-16

**Soundness:** 3
**Presentation:** 3
**Contribution:** 1
**Rating:** 6
**Confidence:** 4

**Summary:**

This paper proposes the in-domain retrieval-augmented framework RAE-XMC for extreme multi-label text classification. RAE-XMC uses a clean contrastive learning function to train and uses a knowledge memory to inference. During inference, the test text retrieves top-b nearest neighbors in the knowledge memory to construct the predicted label.

**Strengths:**

1. The presentation is good.
2. The method is sound.
3. The experimental results are good. In addition, the method converges fastly.

**Weaknesses:**

1. The retrieval-augmented framework for extreme multi-label text classification is not novel. Su et al. (2022) use a similar retrieval-augmented framework for multi-label text classification. The difference is that this paper uses a cleaner contrastive learning loss and incorporates label representations in the memory.

[1] Contrastive learning-enhanced nearest neighbor mechanism for multilabel text classification

**Questions:**

1. It seems that the method has a higher inference overhead. Can you show the inference overhead compared to the OVA and DE methods?
2. Do the methods rely on label descriptions?
3. The improvement mainly comes from the head class in Table 2. Is there some explanation for this? It looks like it's caused by the imbalance in the memory.
4. As shown in Lines 317-323, the method also seems applicable to small-scale multi-label text classifications.

---

> ### Author Response · Authors · 2024-11-24
>
> We sincerely thank the reviewer for their thoughtful and constructive feedback. Below, we address the identified weaknesses and questions in detail.
>
> 1. Comparison with Su et al. (2022):
>     We respectfully disagree with the reviewer’s assessment and provide a different perspective: the RAE-XMC framework is distinct from the method proposed by Su et al. (2022) [1]. While their approach explores a one-versus-all classifier with retrieval-augmented kNN, our method leverages a dual-encoder framework with label text for extreme multi-label classification. This fundamental difference highlights our focus on improving both training efficiency and inference accuracy.
>     Additionally, as shown in Table 1, we directly compare our approach against their method (referred to as OvA+kNN). Our results demonstrate that RAE-XMC significantly outperforms OvA+kNN, primarily due to (1) a more effective training loss and (2) a superior inference strategy that combines kNN predictions with original model predictions. Furthermore, Table 4 highlights the advantage of our inference method over theirs, where ranking instance embeddings and label embeddings within a unified memory outperforms their approach of handling predictions from different sources separately.
>
> 2. Inference Latency:
> OVA methods significantly increase the number of trainable parameters but do not impact inference latency. During inference, both OVA and DE methods follow a similar process: encoding the input into an embedding and retrieving the corresponding label embedding (which can be pre-computed for DE).
>   Our RAE-XMC framework, like other retrieval-augmented methods (e.g., RAG in LLMs), enhances predictive accuracy and training efficiency but incurs a slight increase in inference overhead. We evaluated real-time latency (in milliseconds per query, ms/q) with a batch size of 1, using a single Nvidia A100 GPU for query encoding and a single CPU for approximate nearest neighbor (ANN) search. The results, shown in the table below, indicate that RAE-XMC introduces an additional 2.5 ms/q on average—a reasonable trade-off for the observed performance improvements.
>
> 3. Reliance on Label Descriptions:
>     Dual-encoder-based methods inherently rely on label descriptions. As our method belongs to this category, it also requires label descriptions as an essential component for training and inference.
>
> 4. Head-Class Improvements:
>     As shown in Table 2, when using λ = 1 (where instances are selected and their associated labels are used for prediction), the performance on head labels improves. This behavior aligns with the observation noted in the review: the memory contains a higher proportion of instances associated with head labels, leading to more accurate predictions for these labels.
>     This phenomenon supports our motivation—head labels inherently possess richer information due to their prevalence, but it is challenging for models to memorize all their details effectively. By leveraging memory, RAE-XMC better captures this abundant information, enhancing prediction for head labels.
>     Conversely, the dual-encoder inference approach demonstrates superior performance on tail labels, benefiting from better generalization. This complementary strength between memory-augmented inference for head labels and dual-encoder inference for tail labels underscores the balanced efficacy of our method across the label spectrum.
>
> 5. Applicability to Small-Scale Datasets:
>     While our method is applicable to both small- and large-scale datasets, we emphasize its advantages in addressing the memorization issue prevalent in large-scale datasets. In such cases, the dual-encoder approach reduces the need for the model to memorize every detail, which becomes increasingly challenging with larger datasets. Therefore, our experiments primarily focus on large-scale datasets where these benefits are more evident.
>
> Latency:
> | Methods           | LF-AmazonTitles-131K | LF-WikiSeeAlso-320K | LF-Wikipedia-500K | LF-AmazonTitles-1.3M | Average |
> |--------------------|----------------------|---------------------|--------------------|-----------------------|---------|
> | Dual-Encoder (DE)  | 10.08               | 12.48              | 11.22             | 10.63                | 11.1    |
> | RAE-XMC (ours)     | 12.75               | 13.18              | 14.38             | 13.95                | 13.6    |
>
>
> References:
> [1]Contrastive learning-enhanced nearest neighbor mechanism for multi-label text classification, Su et. al

---

> > ### Comment · Reviewer_FvxB · 2024-11-26
> >
> > Thanks for your response. Although I still have some concerns about the novelty, I raised my score considering the thorough experiments.

---

> > > ### Author Response · Authors · 2024-11-30
> > >
> > > Thank you for your feedback and for considering our response. We appreciate your recognition of our experiments and constructive insights.

---

### Official Review · Reviewer_FsaN · 2024-10-31

**Soundness:** 1
**Presentation:** 3
**Contribution:** 1
**Rating:** 5
**Confidence:** 4

**Summary:**

Paper is about generalization and memorization trade off in extreme classification (XC) and give good quality background and information about recent related works. Proposed approach is scalable and shows state of the art results in XC.

**Strengths:**

-	Very well written, I could follow each and every section
-	Training is scalable and architecture does not add to the training memory

**Weaknesses:**

-	Does not compare with State-of-the-art XC methods like OAK which works in retriever augmented encoders
-	Results are not reported on short text titles datasets, also available on the XC repository. I would suggest authors to report numbers on titles datasets as they are closer to real world tasks.
-	OAK uses auxiliary information, why was this auxiliary information not considered in knowledge memory

**Questions:**

Along with answers to the weakness sections, please address the following questions:

- **The approach is not novel**, many papers in the past have use label centroids (Parabel, Astec, XR-Transformer etc) as well as knn on the training documents (Astec, FasterXML). Since authors cited all of them, Kindly comment on how proposed approach is different for these methods. As of now it looks like authors added KNN over DEXML.

- Since authors main argument is about memorization, authors should also report accuracy of encode on training set. Reason to ask this is, is if accuracy is sufficiently high what is the need of adding knn?

---

> ### Author Response · Authors · 2024-11-24
>
> We sincerely thank the reviewer for their thoughtful and constructive feedback. Below, we address the identified weaknesses and questions in detail.
>
> 1. We appreciate the reviewer’s suggestion and will revise the paper to include the following detailed comparison with OAK:
>     * Settings:
>         OAK utilizes external data, such as Wikipedia articles and related queries generated by ChatGPT-4, which makes the experimental settings fundamentally different. In contrast, the knowledge memory of RAE-XMC only relies on the training split of XMC benchmark datasets. These dependencies on external resources make direct comparisons challenging.
>     * Method:
>         A key distinction lies in how memories are leveraged. OAK combines information from multiple retrieved auxiliary data, irrespective of their associated labels, to form an augmented document representation. In contrast, our method focuses on utilizing the labels of retrieved memories to aid prediction. This difference highlights the complementarity of our methods, suggesting they could be combined for further improvement. Additionally, our method employs a straightforward dual-encoder training setup, while OAK requires a more complex three-stage training pipeline, including a one-vs-all classifier that fine-tunes label embeddings. This results in OAK having a much larger trainable parameters compared to our method.
>     * Performance:
>         Despite not relying on additional data or proprietary LLMs like ChatGPT-4, our simpler approach achieves comparable performance to OAK. The table below highlights performance on two datasets, where our method achieves an average improvement of 0.5% in P@1 and 0.3% in P@5 compared to OAK.
>
> 2. Experiments on Title Datasets:
>     We have conducted experiments on title datasets, as indicated in Table 1. Both LF-AmazonTitles-131K and LF-AmazonTitles-1.3M are title datasets that align closely with real-world tasks.
> 3. Auxiliary Information in Knowledge Memory:
>     In conventional XMC settings, auxiliary information is not provided in the dataset. The auxiliary data used in OAK was manually collected from Wikipedia hyperlinks or generated using proprietary tools like ChatGPT-4. Incorporating such auxiliary information would deviate from standard XMC benchmarks.
> 4. Comparison to Other XMC Methods:
>     The methods mentioned, such as XR-Transformer, Parabel, and FastXML, are primarily tree-based one-vs-all XMC approaches. These methods utilize label centroids and clustering techniques like k-means to construct label trees. Their inference process follows tree paths to retrieve relevant labels, which differs fundamentally from kNN-based retrieval.
>     Compared to DEXML, our method eliminates the need for gradient caching during training, which is essential for DEXML to achieve optimal performance by memorizing all the details in training set. Gradient caching requires computing gradients on the entire dataset, significantly increasing training complexity and cost. During inference, our approach goes beyond simple kNN prediction by combining predictions from both the dual encoder and kNN in an effective and straightforward manner.
> 5. Training Set Performance:
>     The performance on the training set demonstrates that the model cannot fully memorize all training samples. For instance, in the LF-WikiSeeAlso-320K dataset, after 24 training epochs, P@1 only reaches around 82 and saturates with additional training. This indicates that contrastive learning with negative sampling is insufficient for the model to memorize all details. Applying kNN addresses this limitation by reducing the model’s reliance on memorization, thereby improving prediction quality.
>
> Performance comparison with OAK.
> | Dataset | Metric | OAK | RAE-XMC |
> |-------------------------|----------|---------|-----------|
> | LF-WikiSeeAlso-320K | P@1 | 48.57 | 48.04 |
> | | P@5 | 23.28 | 23.68 |
> | LF-Wikipedia-500K | P@1 | 85.23 | 86.49 |
> | | P@5 | 50.79 | 50.67 |

---

> ### Comment · Reviewer_FsaN · 2024-11-25
>
> Dear Authors, I am keeping my scores.
> Reasons are listed as below.
> - Report numbers on all title's dataset, including Wikipedia
> - The auxiliary information is already available as part of raw data dump; Paper is talking about using centroids as additional information why can't that be one used by OAK?
> - Provide a detailed analysis on train set accuracy.
> - Please report numbers on propensity score metric.

---

> ### Author Response · Authors · 2024-12-01
>
> We thank the reviewer for their thoughtful comments and active participation in the discussion. Below, we have carefully addressed the reviewer's concerns:
>
> ### 1. Report numbers on all title's dataset, including Wikipedia
> **Reply**:
> We report the performance on the title dataset of WikiSeeAlso-320K:
>
> | Method  | P@1  | P@5  |
> |-------|-------------|------------|
> |RAE-XMC| 33.08 | 17.08 |
> |OAK| 33.71|17.12|
>
> RAE-XMC achieves similar results compared to OAK, while the latter leverages additional auxiliary data that is beyond the standard training data source of the XMC repo.
>
> ---
>
> ### 2. The auxiliary information is already available as part of the raw data dump. The paper discusses using centroids as additional information. Why can't that be used by OAK?
> **Reply**:
> After carefully reviewing OAK, we found that its auxiliary data was manually curated and is not publicly available. Standard XMC datasets, such as those from the (e.g., [XC XML Repository](http://manikvarma.org/downloads/XC/XMLRepository.html)), commonly used in XMC research, do not include auxiliary data. Furthermore, OAK requires tailored data crafting for each target dataset, which limits its generalizability. For instance, applying OAK directly to datasets like Amazon would require additional auxiliary data mining, which is non-trivial.
>
> ---
>
> ### 3. Provide a detailed analysis on train set accuracy
> **Reply**:
> We evaluated training P@1 at steps 100, 1000, 2000, 3000, and 4000 on WikiSeeAlso-320K. Each epoch corresponds to approximately 100 steps. As shown in the table below, the performance gains in P@1 and P@5 slow down with training progression. This indicates that further increasing training steps alone does not lead to significant improvements.
>
> | Step  | P@1 (Train) | P@1 (Test) | P@5 (Train) | P@5 (Test) |
> |-------|-------------|------------|-------------|------------|
> | 100   | 44.64       | 31.08      | 17.25       | 14.50      |
> | 1000  | 65.63       | 39.28      | 25.08       | 18.86      |
> | 2000  | 80.40       | 42.30      | 29.50       | 19.97      |
> | 3000  | 82.70       | 42.80      | 30.20       | 20.30      |
> | 4000  | 83.50       | 42.90      | 30.40       | 20.30      |
>
> ---
>
> ### 4. Please report numbers on the propensity score metric
> **Reply**:
> Propensity-scored precision (PSP) metrics aim to emphasize the performance on tail labels by assigning higher weights to them. However, as recent work [1][2][3] has shown, PSP metrics are often confounded by the interplay between tail label performance and label missingness, making them an unreliable indicator of tail label performance. Following the recommendation of [1], we reported the macro-average F1 score in Table 2 as a more robust metric.
>
> References:
> [1]Generalized test utilities for long-tail performance in extreme multi-label classification, Schultheis et. al.
> [2]On Missing Labels, Long-tails and Propensities in Extreme Multi-label Classification, Schultheis et. al.
> [3]Long-tailed Extreme Multi-label Text Classification by the Retrieval of Generated Pseudo Label Descriptions, Zhang et. al.

---

> > ### Comment · Reviewer_FsaN · 2024-12-02
> >
> > Dear Authors,
> > - Thank you for providing numbers on titles datasets, since numbers on titles datasets are low, I believe much works needs to be done on the algorithm.
> > - Having task specific metadata is precisely the point of OAK. I am not sure why having centroid-based metadata is more critical.
> > - Please focus on P@1 for training, for WikiSeeAlso dataset, average number of labels per datapoints are 2-3 only. Therefore, tracking P@5 for early stopping is not reliable. I believe authors should spend more time on the analysis.
> > - PSP point well taken.
> >
> > I have increased my score.

---

### Official Review · Reviewer_sr4q · 2024-11-03

**Soundness:** 3
**Presentation:** 4
**Contribution:** 2
**Rating:** 5
**Confidence:** 5

**Summary:**

The Retrieval-augmented Encoders for XMC (RAE-XMC) framework enhances dual-encoder (DE) models with retrieval capabilities, improving memorization without adding extra trainable parameters. RAE-XMC uses contrastive loss over a knowledge memory of input instances and labels during training. For inference, it retrieves top-K keys from this memory and aggregates their values for prediction scores. Demonstrated on four public LF-XMC benchmarks, RAE-XMC surpasses the state-of-the-art method DEXML, achieving over 10x speedup on the LF-AmazonTitles-1.3M dataset using the same 8 A100 GPUs training setup.

**Strengths:**

1. The idea of using retrieval augmentation with XMC is very interesting.
2. Storing existing dataset samples in memory is a nice trick.
3. Training seems to be very efficient. Also, the authors have performed extensive experimentation under many settings.

**Weaknesses:**

1. OAK is a recent method which also uses memory for XMC tasks. How does RAE-XMC compare with OAK?
2. PSP metrics seem to be very commonly used across XMC literature. Can you please report PSP also?
3. Table 1: Does TT include memory construction time also? If not, it will be nice to include that also.
4. Improvements are somewhat weak in Table 1. On LF-AmazonTitles-131K, P@1 is not best for RAE-XMC. On LF-WikiSeeAlso-320K it looks like RAE-XMC is not stat sig better than NGAME. Also, on LF-AmazonTitles-1.3M, RAE-XMC is not stat sig better than DEXML.
5. Since memory needs to be stored as well, how do these methods compare with respect to their RAM requirements?
6. From a novelty perspective, the method looks like Approximate KNN using encoders (trained with std methods taken from DEXML and NGAME). Although it is being called called retrieval augmented, there is actually no augmentation here at all. It is more of KNN rather than retrieval augmentation.
7. Case studies: It would be nice to see top b retrieved samples for some sample, and show why those help to improve accuracy of the proposed method compared to DEXML.
8. Also some error analysis would be nice to do, especially on samples where RAE-XMC was wrong but DEXML was correct. What percent of errors are because of wrong top b neighbors?
9. Can lambda be learned/tuned/computed per sample?
10. Line 192: what is f?

**Questions:**

Please see weaknesses

---

> ### Author Response · Authors · 2024-11-24
>
> We sincerely thank the reviewer for their thoughtful and constructive feedback. Below, we address the identified weaknesses and questions in detail.
>
> 1. We appreciate the reviewer’s suggestion and will revise the paper to include the following detailed comparison with OAK:
>     * Settings:
>         OAK utilizes external data, such as Wikipedia articles and related queries generated by ChatGPT-4, which makes the experimental settings fundamentally different. In contrast, the knowledge memory of RAE-XMC only relies on the training split of XMC benchmark datasets. These dependencies on external resources make direct comparisons challenging.
>     * Method:
>         A key distinction lies in how memories are leveraged. OAK combines information from multiple retrieved auxiliary data, irrespective of their associated labels, to form an augmented document representation. In contrast, our method focuses on utilizing the labels of retrieved memories to aid prediction. This difference highlights the complementarity of our methods, suggesting they could be combined for further improvement. Additionally, our method employs a straightforward dual-encoder training setup, while OAK requires a more complex three-stage training pipeline, including a one-vs-all classifier that fine-tunes label embeddings. This results in OAK having a much larger trainable parameters compared to our method.
>     * Performance:
>         Despite not relying on additional data or proprietary LLMs like ChatGPT-4, our simpler approach achieves comparable performance to OAK. The table below highlights performance on two datasets, where our method achieves an average improvement of 0.5% in P@1 and 0.3% in P@5 compared to OAK.
>
> 2. Report PSP:
> Propensity-scored precision (PSP) metrics aim to emphasize the performance on tail labels by assigning higher weights to them. However, as recent work [1][2][3] has shown, PSP metrics are often confounded by the interplay between tail label performance and label missingness, making them an unreliable indicator of tail label performance. Following the recommendation of [1], we reported the macro-average F1 score in Table 2 as a more robust metric.
>
> 3. Memory Construction Time:
> Memory construction in our method refers to encoding texts into embeddings, which is performed once and does not occur during training. Therefore, we excluded it from TT. The memory construction time is minimal, accounting for less than 1.5% of the total training time. For example, in the AmazonTitles-131K dataset, memory construction took only 35 seconds compared to 36 minutes for model training.
>
> 4. Improvements and Novelty:
> Our method is designed to simplify and accelerate the training of pure dual-encoder models. rather than solely maximizing performance. By retrieving relevant documents and their associated labels, our approach reduces the need for XMC models to memorize the entire dataset, resulting in a more efficient training process. For example, on the largest LF-AmazonTitles-1.3M dataset, RAE-XMC speedup 10x compared to DEXML, the SOTA dual-encoder model.
>
> 5. Can lambda be learned/tuned/computed per sample?
> Thanks for the suggestion, we will leave data-driven or trainable lambda as the future work.
>
> 6. Can lambda be learned/tuned/computed per sample?
> Thank you for the suggestion. Exploring data-driven or trainable lambda is an excellent direction, and we plan to investigate it as future work.
>
> 7. Line 192: what is f:
>     We appreciate the observation. The function f is defined in Section 2, "Background Material", as a text encoder. To improve clarity, we will also provide a brief explanation of at Line 192.
>
>
> Performance comparison with OAK.
> | Dataset | Metric | OAK | RAE-XMC |
> |-------------------------|----------|---------|-----------|
> | LF-WikiSeeAlso-320K | P@1 | 48.57 | 48.04 |
> | | P@5 | 23.28 | 23.68 |
> | LF-Wikipedia-500K | P@1 | 85.23 | 86.49 |
> | | P@5 | 50.79 | 50.67 |
>
> References:
> [1]Generalized test utilities for long-tail performance in extreme multi-label classification, Schultheis et. al.
> [2]On Missing Labels, Long-tails and Propensities in Extreme Multi-label Classification, Schultheis et. al.
> [3]Long-tailed Extreme Multi-label Text Classification by the Retrieval of Generated Pseudo Label Descriptions, Zhang et. al.

---

> > ### Comment · Reviewer_sr4q · 2024-11-25
> >
> > Thank you for your response. Thanks for comparing with OAK even though it uses extra signals.
> >
> > My concerns about lack of deep novelty, minor improvements over SOTA, lack of error analysis, case studies still remain. Hence, I will keep my score.

---

> ### Author Response · Authors · 2024-11-30
> **Case Study**
>
> We thank the reviewer for their thoughtful comments and active participation in the discussion. To address the concerns raised, we have conducted a case study, which is detailed below:
>
> In the WikiSeeAlso-320K dataset, for the input article "History of Guyana," RAE-XMC retrieves both articles, such as "History of Suriname" and "President of Guyana," as well as labels like "Republic of Independent Guyana," "Culture of Guyana," and "History of South America." Suriname, being a neighboring country with a shared historical context, has overlapping "See Also" labels with Guyana. As a result, retrieving the article "History of Suriname" helps RAE-XMC predict correct labels such as "History of South America," "History of the Americas," "Spanish colonization of the Americas," and "French colonization of the Americas."
>
> In contrast, DEXML retrieves labels such as "History of South America," "List of heads of state of Guyana," and "Republic of Independent Guyana." While some of these labels are accurate, many are only loosely related to "Guyana" rather than specifically focusing on its history. This highlights RAE-XMC's strength in retrieving contextually relevant labels, often resulting in more comprehensive predictions, despite some remaining gaps.
>
> | Input Article Title                    | History of Guyana    |
> |---------------------------------------------------------------------------------|-----------------------------------------------------------------------------------------------------------------------------------------------------------------------------------------------------------------------------------------------|
> | Input Article  Content  | The recorded history of Guyana can be dated back to 1466, when Guyana de Ojeda's first expedition arrived from Spain at the Essequibo River...                                                                                                  |
> | Input Article Labels             | History of the Americas, History of South America, Spanish colonization of the Americas, History of the Caribbean, British colonization of the Americas, French colonization of the Americas, History of the British West Indies, Politics of Guyana, List of Governors of British Guiana, List of Prime Ministers of Guyana |
> | Retrieved top-5 Texts Ordered by Score (RAE-XMC)      | **1. Label:** "Republic of Independent Guyana". **2. Article Title:** "History of Suriname"  Article Labels: History of the Americas, History of South America, Spanish colonization of the Americas, History of the Caribbean, British colonization of the Americas, French colonization of the Americas, Dutch colonization of the Americas, Politics of Suriname, History of Suriname **3. Article Title:** "President of Guyana" Article Labels: List of Governors of British Guiana, List of Prime Ministers of Guyana, List of heads of state of Guyana **4. Label:** "Culture of Guyana" **5. Label:** "History of South America" |
> | Predicted Labels (RAE-XMC)       | History of South America, Spanish colonization of the Americas, History of the Americas, British colonization of the Americas, French colonization of the Americas                                                 |
> | Predicted Labels (DEXML)         | Republic of Independent Guyana, History of South America, List of heads of state of Guyana, History of Suriname, List of Governors of British Guiana, List of Prime Ministers of Guyana          |

---

> > ### Author Response · Authors · 2024-11-30
> > **Error Analysis on WikiSeeAlso-320K**
> >
> > RAE-XMC occasionally retrieves articles with similar literal meanings but unrelated content, leading to prediction errors. Our analysis shows that among the top 5 retrieved articles, 70% are irrelevant to the input text (i.e., they share no labels with the input). However, leveraging Eq. (5), we aggregate scores from the top 200 retrieved labels or articles. This approach mitigates the impact of irrelevant articles, as their scores are dispersed, while relevant articles have more concentrated scores. As shown in Appendix Figure 3 (c) and (d), increasing the number of retrieved texts significantly enhances performance.
> >
> > For instance, consider the example below. The input article discusses “Moraine”—a glacially formed accumulation of debris—while the retrieved article focuses on “Moraine Lake”, a lake in Canada. This mismatch causes RAE-XMC to underperform compared to DEXML, predicting labels such as “Natural environment” and “List of lakes” based on the retrieved article.
> >
> > | **Input Article Title**     | Moraine                                                                                                                                                                                                |
> > |-----------------------------|---------------------------------------------------------------------------------------------------------------------------------------------------------------------------------------------------------|
> > | **Input Article Description** | A moraine is any glacially formed accumulation of unconsolidated glacial debris (regolith and rock) that occurs in both currently and formerly glaciated regions on Earth...                             |
> > | **Input Article Labels**    | Valparaiso Moraine, Glacial landform, Drumlin, Esker, Terminal moraine, Dogger Bank, Long Island, Oak Ridges Moraine, Cypress Hills (Canada)                                                            |
> > | **RAE-XMC Top-1 Retrieved Article Title** | Moraine Lake                                                                                                                                                                                       |
> > | **RAE-XMC Top-1 Retrieved Article Content** | Moraine Lake is a glacially fed lake in Banff National Park, outside the Village of Lake Louise, Alberta, Canada.                                                                                   |
> > | **RAE-XMC Top-1 Retrieved Article Labels** | Natural environment, List of lakes, Nature                                                                                                                                                             |
> > | **Predicted Labels (RAE-XMC)** | List of glacial moraines, Terminal moraine, Pulju moraine, Natural environment, List of lakes                                                                                                         |
> > | **Predicted Labels (DEXML)** | Terminal moraine, Valparaiso Moraine, List of glacial moraines, Mississinawa Moraine, Glacial landform                                                                                                  |
> >
> > This analysis highlights a common challenge for RAE-XMC: retrieving articles with semantically related but contextually mismatched content. While score aggregation reduces the impact of irrelevant articles, further improvements are needed to enhance retrieval precision and contextual alignment.

---

### Official Review · Reviewer_kCmc · 2024-11-06

**Soundness:** 2
**Presentation:** 3
**Contribution:** 2
**Rating:** 5
**Confidence:** 4

**Summary:**

This paper studies the Extreme multi-label classification problem via proposing the Retrieval-augmented Encoders for XMC (RAE-XMC) framework. This framework equips a DE model with retrieval-augmented capability for efficient memorization. The empirical study confirms the effectiveness and efficiency of four public benchmarks. In addition, the authors shows the presented approach achieves significant speedup on the largest dataset.

**Strengths:**

1. This study provides a comprehensive overview and in-depth summarization of the various existing approaches that have been developed for extreme multi-label classification. A thorough analysis is conducted on the advantages and disadvantages of each type of approaches. method. This ensures that readers gain a robust understanding of the current models available.

2. The concept of introducing the retrieval-augmented method is interesting. This presents interesting possibilities for improving the performance of the extreme multi-label classification tasks.

**Weaknesses:**

1. My primary concern regarding this study is on the aspect of novelty. The concept of incorporating retrieval-augmented knowledge certainly has the potential to provide valuable background information that can enhance classification performance. However, aside from this innovative idea, the overall design of the model remains quite conventional and adheres to traditional methodologies, which may limit its effectiveness.

2. There is a lack of detailed information regarding the implementation of this study. Furthermore, it could be beneficial to explore the potential of utilizing various other language models or, ideally, open-source large language models to assess the feasibility of integrating the proposed model with those advanced technologies. This exploration could offer insights into how to enhance the performance and capabilities of the existing framework.

**Questions:**

Please see Weaknesses.

---

> ### Author Response · Authors · 2024-11-24
>
> We sincerely thank the reviewer for their thoughtful and constructive feedback. Below, we address the identified weaknesses and questions in detail.
> 1. Our primary objective is to improve training efficiency of dual-encoder models for XMC problem. For the largest LF-AmazonTitles-1.3M dataset, RAE-XMC speedup 10x compared to DEXML, the SOTA dual-encoder model. We selected DistilBERT as our backbone model because of its smaller size, which aligns with our goal of efficiency. Despite its simplicity, our method achieves state-of-the-art results by reducing the reliance of XMC models on memorizing the entire dataset. This demonstrates that even a lightweight approach can yield highly competitive performance when paired with our framework.
> 2. Regarding implementation, we will release our code after the anonymous review period. Our proposed method is designed to be adaptable to any advanced encoder without requiring fine-tuning, including instruction-tuned large language models. Table 5 illustrates this adaptability by showing the performance of GTE-base (https://huggingface.co/thenlper/gte-base) within our RAE-XMC inference framework, used without any encoder training. The results confirm that using more advanced encoders substantially improves performance compared to the DistilBERT encoder. This suggests that further improvements are achievable with more sophisticated encoder architectures.

---

### Meta-Review · Area_Chair_cKNz · 2024-12-26

**Metareview:**

All reviewers find the paper to be well written. However they found proposed approach not novel and advantages of the considered approach to be limited. Authors provided additional comparisons with other approaches during the discussion phase. However this didnt alleviate the reviewer concerns as the gains seemed not sufficient. Overall substantial revision is required to accept the paper.

**Additional Comments On Reviewer Discussion:**

Reviewers find the approach to be limited with missing comparisons. Authors provided additional results, but the overall contributions seem limited to the reviewers.

---

### Decision · Program_Chairs · 2025-01-22

Reject